# Anomalous trends in global ocean carbon concentrations following the 2022 eruptions of Hunga Tonga-Hunga Ha'apai
Bryan A. Franz [1] ✉, Ivona Cetinić [1,2] ✉, Amir Ibrahim[1] ✉ & Andrew M. Sayer [1,3] ✉

We report on observed trend anomalies in climate-relevant global ocean biogeochemical properties, as derived from satellite ocean color measurements, that show a substantial decline in phytoplankton carbon concentrations following eruptions of the submarine volcano Hunga Tonga-Hunga Ha'apai in January 2022. The anomalies are seen in remotely-sensed ocean color data sets from multiple satellite missions, but not in situ observations, thus suggesting that the observed anomalies are a result of ocean color retrieval errors rather than indicators of a major shift in phytoplankton carbon concentrations. The enhanced concentration of aerosols in the stratosphere following the eruptions results in a violation of some fundamental assumptions in the processing algorithms used to obtain marine biogeochemical properties from satellite radiometric observations, and it is demonstrated through radiative transfer simulations that this is the likely cause of the anomalous trends. We note that any future stratospheric aerosol disturbances, either natural or geoengineered, may lead to similar artifacts in satellite ocean color and other remote-sensing measurements of the marine environment, thus confounding our ability to track the impact of such events on ocean ecosystems.

A recent analysis of the global satellite ocean color (OC) trends from NASA's Moderate Resolution Imaging Spectroradiometer (MODIS) on Aqua revealed peculiar patterns in derived concentrations of the phytoplankton pigment chlorophyll-a and phytoplankton carbon for the year 2022, with annual trends in phytoplankton carbon concentrations indicating behavior well outside of the 25 yr climatological record[1]. The anomalies in global phytoplankton metrics were driven by deviations in the equatorial region (23.5° N to 23.5° S) and the southern hemisphere. This analysis did not detect visible impact to the OC products from the northern hemisphere, and speculated that the unusual trends were likely due to recent volcanic eruptions near Tonga[1].

On the 13th and 15th of January 2022, two major eruptions of the submarine volcano Hunga Tonga-Hunga Ha'apai (located at 20.55°S, 175.4°W) injected an unprecedented amount of water vapor (estimated $146 \pm 5$ Tg) and a modest amount of $SO_2$ (estimated ~0.4 Tg) into the stratosphere[2–5]. Carr et al. reported visible plumes (via stereo imaging) to a record height of 50–55 km[6]; Legras et al. (2022) observed oxidation of this injected $SO_2$ to sulfate aerosols over several weeks, and the formation of aerosol layers from 24–26 km[7]. While gradually declining in concentration and altitude, elevated stratospheric aerosol loading persisted throughout 2022 and beyond[4], and the $H_2O$ enhancement is expected to persist for some time as well[2].

While the atmospheric and radiative impacts of the eruptions have been discussed[8], here we focus on reporting the impact of the eruption on ocean color products retrieved from a host of spaceborne spectral radiometers measuring in the visible to near-infrared spectral range. These satellite sensors are designed to measure variations in the bio-optical and biogeochemical properties of the global oceans, including distributions of the phytoplankton pigment chlorophyll-a (Chla; mg m$^{-3}$) or concentration of phytoplankton carbon (Cphy; mg m$^{-3}$), which can be derived from the particulate backscattering coefficient of seawater ($b_{bp}$(443); m$^{-1}$). These properties are inferred from measurements of ocean color, as characterized through spectral variations in remote sensing reflectance (Rrs($\lambda$); sr$^{-1}$): the radiometric signal upwelling from beneath the ocean surface and measured just above the surface at multiple sensor wavelengths, $\lambda$, in the visible spectral range. The retrieval of Rrs($\lambda$) from satellite sensor radiometry requires a correction for the effects of the atmosphere on the signal received at the sensor. The atmospheric correction (AC) process accounts for the effects of scattering and absorption by atmospheric aerosols, as well as Rayleigh scattering by air molecules and transmittance losses due to absorbing atmospheric gases[9–11].

[1]NASA Goddard Space Flight Center, Greenbelt, MD, USA. [2]Morgan State University, Baltimore, MD, USA. [3]University of Maryland Baltimore County, Baltimore, MD, USA. ✉e-mail: bryan.a.franz@nasa.gov; ivona.cetinic@nasa.gov; amir.ibrahim@nasa.gov; andrew.sayer@nasa.gov

In this study, we provide a detailed assessment of the impact of the Tonga eruptions on satellite OC retrievals through analysis of the OC time series from multiple satellite missions processed by both NASA and ESA/EUMETSAT, and contrast that analysis with time series of in situ measurements from bio-optical floats. We conclude that the primary impact of the eruptions is to introduce systematic error in the AC process, leading to erroneous spatiotemporal anomalies that could be misinterpreted as indicators of biological response. Furthermore, through simulation studies we investigate how the addition of stratospheric aerosols propagates through the NASA standard AC algorithm to produce the observed anomalies in the OC time series and thereby reveal a potential path to mitigation.

## Results and discussion
### Identification of anomalous trends
Satellite remote sensing of ocean color, as characterized by Rrs(λ), provides critical information to support global and regional-scale research and applications on biological and biogeochemical constituents of the world's oceans, including assessment of large-scale changes in the distribution of marine phytoplankton due to natural or anthropogenically-induced

variations in the Earth's climate[12,13]. The capability to track such climate-driven impacts is paramount, as ocean phytoplankton are responsible for roughly 50% of global net primary production, and rapid variations in phytoplankton populations can dramatically alter ocean ecosystems and the services those ecosystems provide, including impacts to food security and global biogeochemical cycles[14,15]. In open oceans, temporal variations in the ocean color signal over a fixed region typically demonstrate a clear seasonal cycle due to natural variations in light and nutrient availability that drive phytoplankton productivity, with deviations from that climatological cycle attributable to regional or global disturbances to the environment that impact phytoplankton growth.

Analysis of the MODIS Aqua OC data record for two key wavelengths in in the blue and green spectral region (Rrs at 443 nm and 547 nm, respectively), and two OC-derived phytoplankton biomass metrics, $b_{bp}(443)$ (directly related to phytoplankton carbon biomass) and Chla, for the NH and SH regions, demonstrates the distinct seasonal cycles in phytoplankton growth that follow the variation in solar illumination for the period of 2002–2021 (Fig. 1). In 2022, however, the seasonal cycle in Rrs(λ) for the SH (red lines in Fig. 1) indicates a strong negative deviation from that 20 year

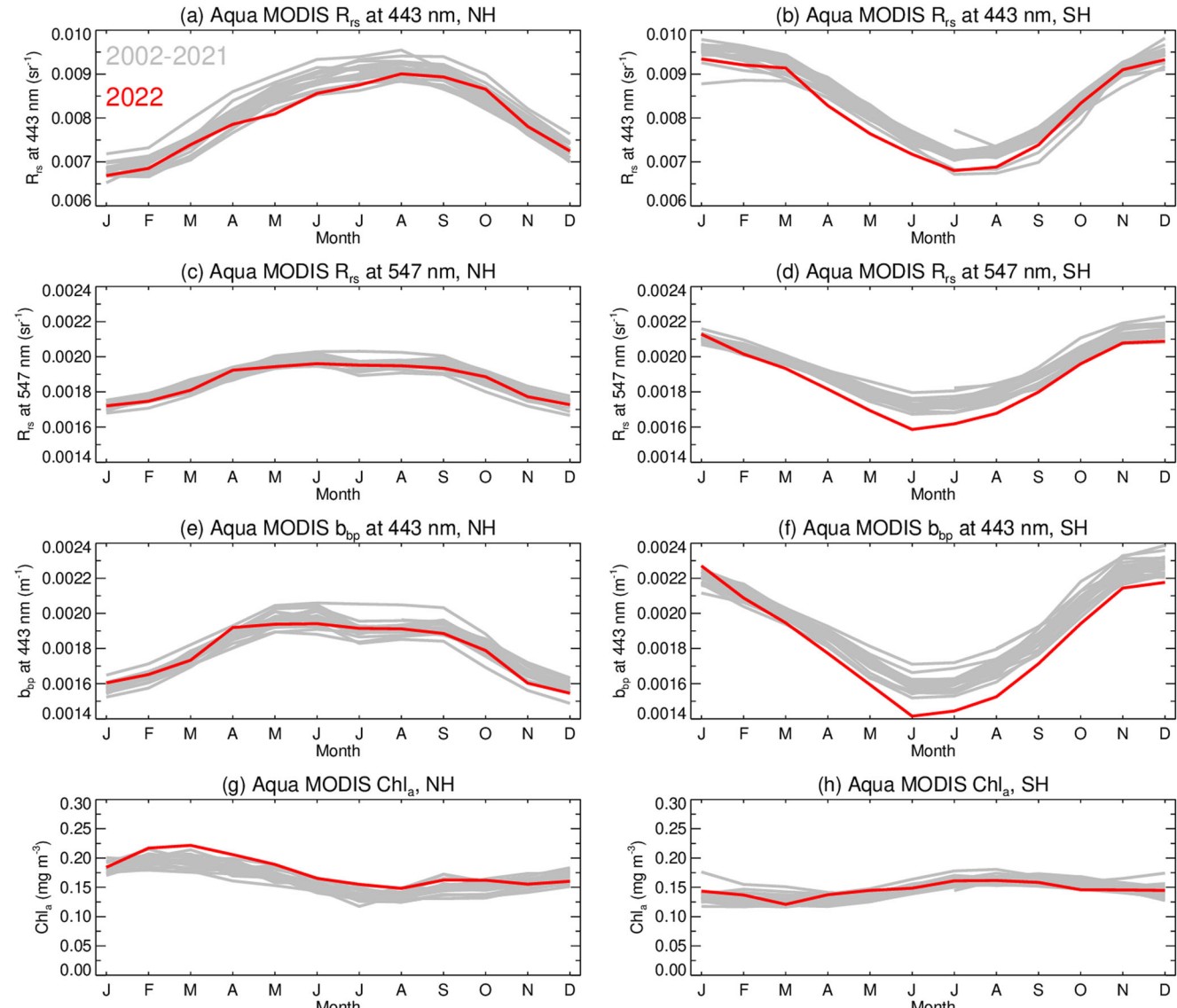

**Fig. 1 | Interannual variability of Ocean color (OC) parameters from MODIS Aqua.** Variability of the OC parameters is depicted for the area of the Northern Hemisphere, 0–50° N (**a, c, e,** and **g**), and Southern Hemisphere, 0–50° S (**b, d, f,** and **h**). Lines depict data for 2022 (red) compared to 2002–2021 (gray). Panels show (**a, b**) Rrs at 443 nm; (**c, d**) Rrs at 547 nm; (**e, f**) $b_{bp}$ at 443 nm; and (**g, h**) derived Chla concentration.

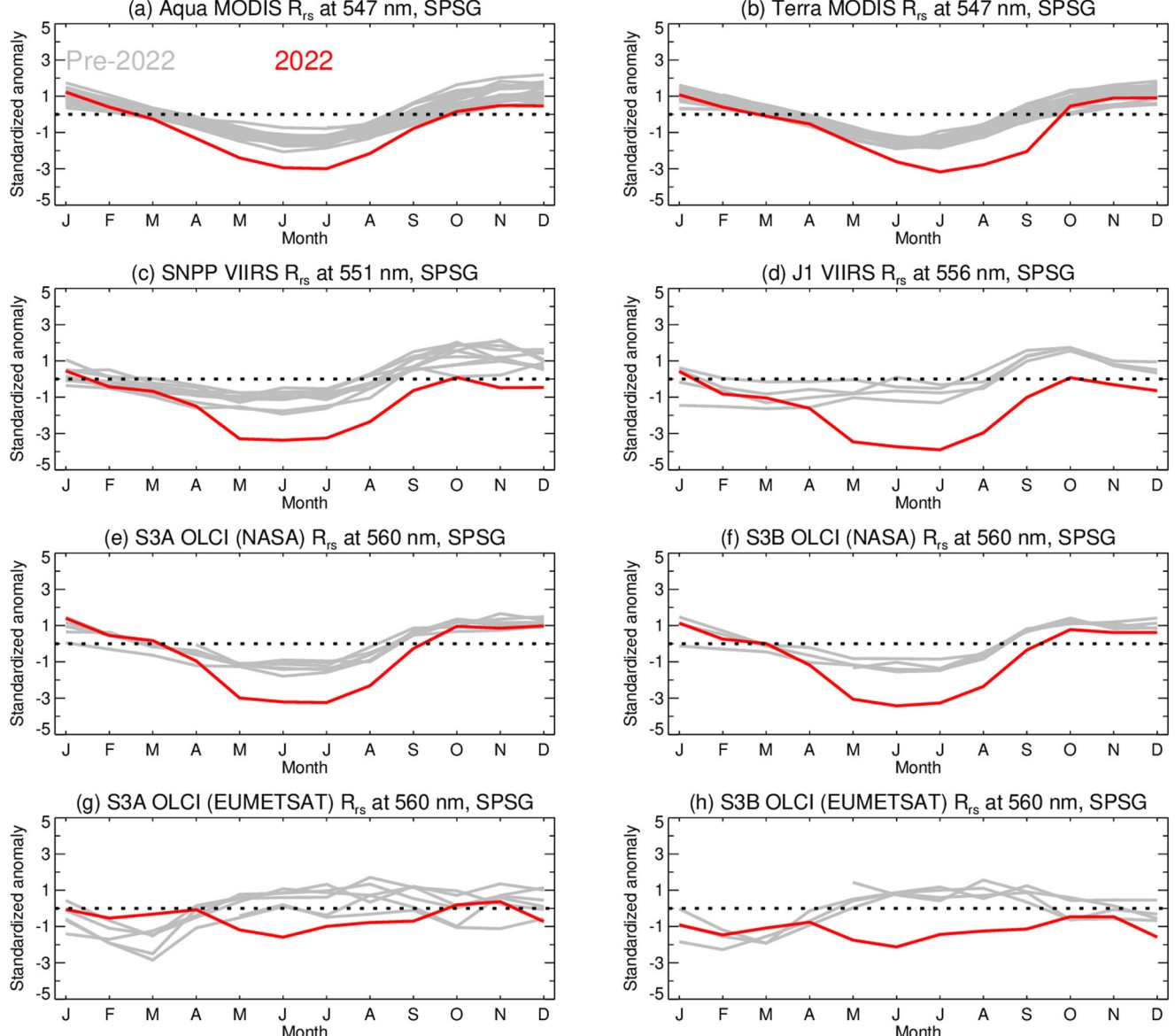

**Fig. 2 | Interannual variability of remote sensing reflectance (Rrs) at green wavelengths in the South Pacific Subtropical Gyre (SPSG) from various satellite sensors.** Data expressed as standardized anomalies relative to the pre-2022 multi-annual mean for the sensor (see Methods). Solid lines show data for 2022 (red) compared to previous years (gray); the black dashed line indicates zero. Exact wavelengths vary from sensor to sensor. Panels show (**a**, **b**) MODIS on Aqua (2002–2022) and Terra (2000–2022); (**c**, **d**) VIIRS on SNPP (2012–2022) and JPSS-1 (aka NOAA-20) (2018–2022); (**e**, **f**) OLCI on Sentinel 3a (2016–2022) and 3b (2018–2022) processed with the NASA algorithms; and (**g**, **h**) the same OLCI measurements as (**e**, **f**) but processed with EUMETSAT algorithms, respectively.

record, with larger deviation in the green (547 nm) than in the blue. A similar discrepancy is visible in the $b_{bp}(443)$ seasonal cycle in SH, while $b_{bp}(443)$ and $Rrs(\lambda)$ in the NH are consistent with the historical norm. In contrast, the Chla seasonal cycle for 2022 is in-family with the climatological record for the SH region and is slightly elevated over much of the year for the NH, consistent with expectations due to the persistent La Niña conditions prevailing in 2022[1].

The same anomalous behavior in ocean color measurements is visible across several other satellite sensors operational in 2022, as shown here for Rrs in the green spectral range, for the SPSG region of the SH (Fig. 2). This cross-sensor comparison demonstrates that the year 2022 was consistently anomalous relative to the climatological record, with deviations typically around 3 standard deviations below the long-term mean for each sensor. We note that, while there is some temporal cross-calibration applied between MODIS-Terra and MODIS-Aqua[16], all other sensors are fully independent with respect to calibration, thus ruling out the possibility that observed anomalies in the SH (Fig. 1) are due to systematic error in the MODIS-Aqua

calibration related to uncorrected radiometric degradation. We note also that similar deviations are visible in datasets independent of the NASA atmospheric correction process. Seasonal cycles in the EUMETSAT-derived Rrs from the Ocean and Land Color Instrument (OLCI), while flatter relative to the NASA-derived OLCI Rrs of Fig. 2e, f, clearly demonstrate the year 2022 to be out of family and with a negative deviation relative to their historical record (Fig. 2g, h).

### Explanation for anomalous trends
The observed rapid change in SH ocean color parameters in 2022 coincided with the eruptions of Hunga Tonga-Hunga Ha'apai, which had an unprecedented impact on the concentration and distribution of atmospheric aerosols. The annual progression of stratospheric aerosol, as measured by NASA's Ozone Mapping and Profiler Suite (OMPS) over 2022, differs substantially from the previous 10 years of measurements (Fig. 3). Near the equator and extending well into the SH, the stratospheric aerosol optical

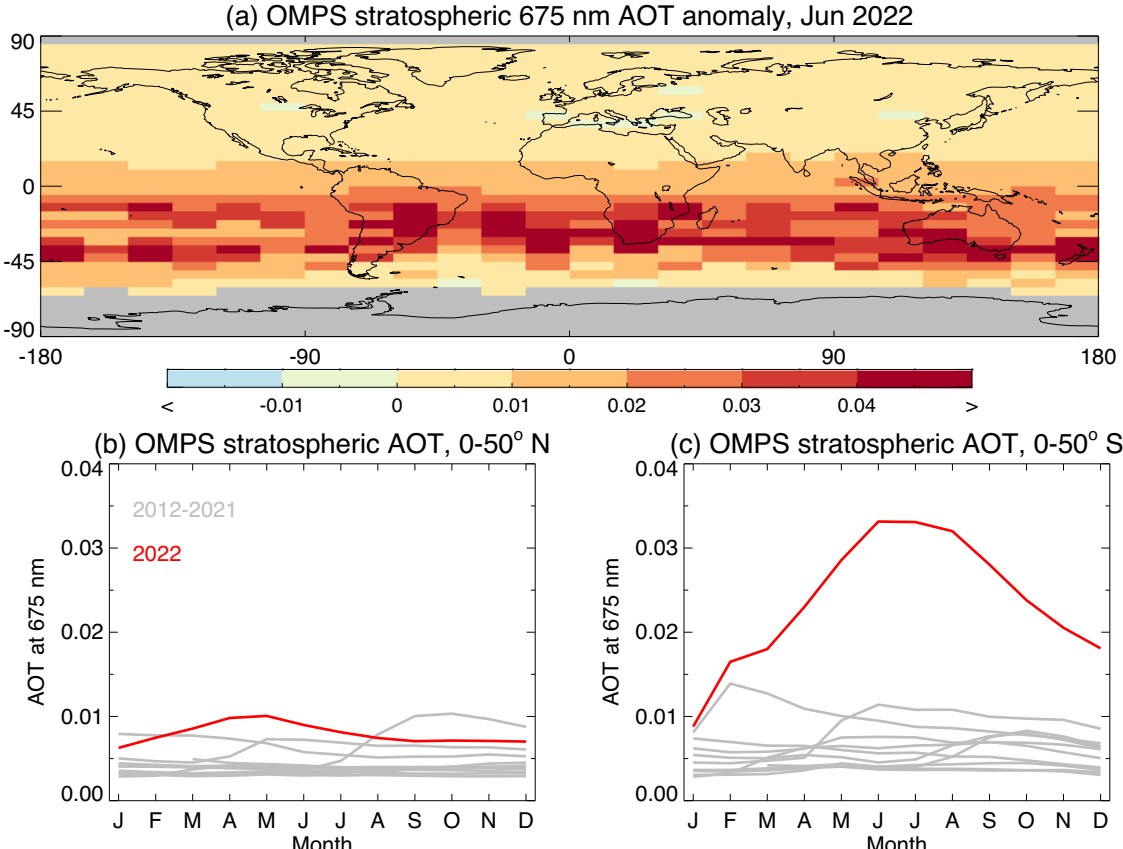

**Fig. 3 | OMPS stratospheric Aerosol Optical Thickness (AOT) at 675 nm during 2022. a** Shows the anomaly for June 2022 relative to the months of June for the 2012–2021 period. Grid cells with fewer than 6 months over this period are shaded gray. **b, c** Depict annual variation of area-weighted geometric mean stratospheric AOT for 2022 (red) compared to 2012–2021 period (gray), for NH (0–50° N) and SH (0–50° N), respectively.

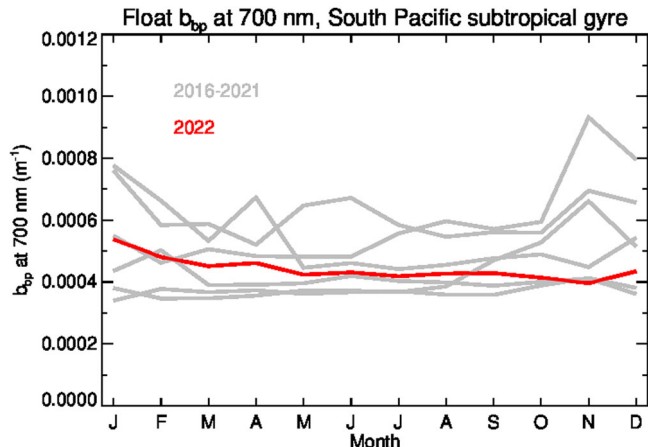

**Fig. 4 | Interannual variability of particulate backscatter at 700 nm, $b_{bp}(700)$, in the South Pacific Subtropical Gyre (SPSG) as measured in situ from BGC-Argo floats.** Red line depicts the average value of all measurements in SPSG during 2022, while gray lines indicate same values for the years during the 2016–2021 period.

thickness (AOT) reported at 675 nm increased dramatically, reaching a peak in June-July of 2022 that was 6–7 times the historical average. The delayed peak in AOT is due to the oxidization process that gradually converted the initial injection of $SO_2$ into stratospheric sulfate aerosols[7]. This progression to positive anomaly in the SH stratospheric aerosol follows closely with the progression of negative $Rrs(\lambda)$ anomalies observed in NASA and ESA/EUMETSAT OC data (Figs. 1, 2).

The anticorrelation in ocean color (Figs. 1, 2) and aerosol trends (Fig. 3) over the year 2022 suggests that there is likely a causal link between the unusual distribution of aerosols from the Tonga eruptions and the OC SH anomalies, drawing two possible hypotheses. The first hypothesis is that we are observing a biological response of the phytoplankton community to the eruption, as has been observed in previous volcanic events and linked to transport of volcanic ash, which typically contains nutrients such as iron needed for photosynthesis, to nutrient deplete regions of the oceans[17–19]. What we observe in the ocean color trends for 2022 in the SH is a marked decrease in $b_{bp}$ paired with minimal change in chlorophyll concentration, which could be interpreted as a physiological response of the phytoplankton community that resulted in a decrease in phytoplankton biomass (Cphy). The second hypothesis is that the anomalies observed in the satellite ocean color record are the result of an error in the atmospheric correction process due to the unusual aerosol conditions following the eruption. To evaluate these hypotheses, an independent dataset of phytoplankton biomass collected in situ by the BGC-Argo fleet was assessed (Fig. 4). The BGC-Argo measurements of $b_{bp}$ from within the SPSG indicate no change in 2022 relative to previous years (2016–2022). This supports the assertion in Franz et al. that the anomalies observed in the satellite ocean color record are likely the result of error in the atmospheric correction process rather than an indication of a marked decrease in phytoplankton biomass in 2022[1].

The NASA AC algorithm assumes that aerosols are primarily scattering, with only weak absorption, and that those aerosols are located in the troposphere[10]. When first injected, the additional aerosols from the Tonga eruptions are believed to have been moderately absorbing[20], but after aging and transport, the sustained anomalous aerosol population in the stratosphere is found to be weakly absorbing[8], consistent with the NASA AC algorithm assumptions. The AC algorithm also assumes that the aerosol

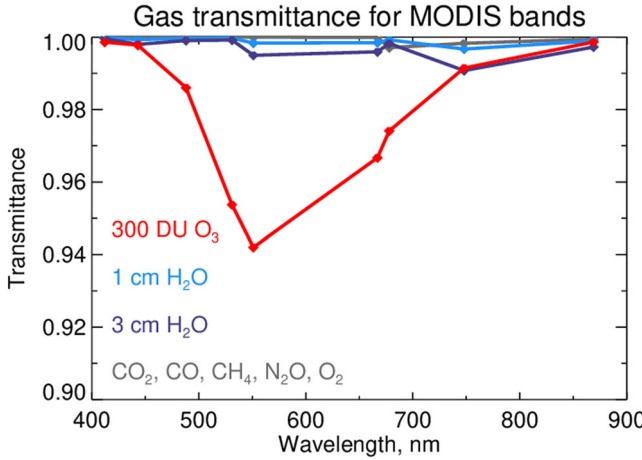

**Fig. 5 | Transmittance losses due to various atmospheric gases within the MODIS spectral bands used for ocean color retrieval.** These values were derived by integration of the associated MODIS Aqua spectral response functions with the absorbing gas spectra, where integration of different gases is indicated by different color lines: 300 DU $O_3$ ozone (red); 1 cm $H_2O$—water vapor (light blue); 3 cm $H_2O$—water vapor (dark blue); $CO_2$, $CO, CH_2$, $N_2O$, $O_2$—other gasses (in gray).

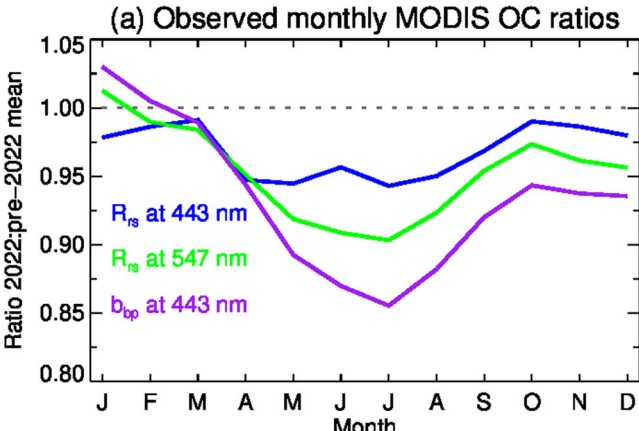

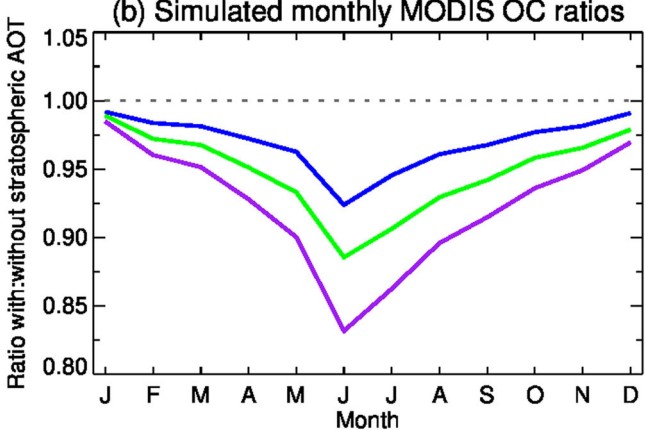

**Fig. 6 | Sensitivity analysis of the impact of stratospheric aerosols on OC product retrievals following the Hunga Tonga-Hunga Ha'apai eruption. a** Shows the ratio of monthly mean OC product retrievals for 2022 relative to the monthly mean of 2003–2021, as derived from MODIS Aqua measurements of the SPSG region. **b** Shows the ratio of monthly OC product retrievals as derived from simulated MODIS Aqua observations, with stratospheric aerosols included, relative to simulation and retrieval with no stratospheric aerosols included. Remote sensing reflectance in blue (Rrs at 443 nm) is depicted with blue line, Remote sensing reflectance in green (Rrs at 547 nm) is depicted with blue line, and backscattering ($b_{bp}$ at 443 nm) is depicted with purple line. See text for a detailed description of the simulation assumptions.

(and most of the atmosphere) is below the stratospheric ozone layer, and therefore a correction for ozone absorption can be applied before attempting to separate the atmosphere and ocean contributions from the total signal reaching the satellite sensor[10]. The additional aerosols from the Tonga eruptions, however, were injected into the stratosphere and thus co-mingled with the ozone, breaking an inherent assumption in the NASA AC algorithm. Furthermore, while the spectral band locations and band widths used for ocean color retrievals are designed to minimize sensitivity to absorbing atmospheric gases, such as water vapor and oxygen, the broad-band absorption spectrum of ozone cannot be avoided, and thus ocean color retrievals are sensitive to ozone absorption, with strongest sensitivity in the 500–700 nm range (Fig. 5).

To evaluate the second hypothesis, we assess the impact of this unusual aerosol-ozone mixing on the AC process, and consequently on the derived OC products, through a sensitivity analysis that was conducted using a fully-coupled ocean-atmosphere vector radiative transfer simulation (Fig. 6). This simulation incorporated atmospheric conditions indicative of post-eruption scenarios, with stratospheric aerosol distributions based on Taha et al.[4]. We simulated total reflectance at the satellite sensor, both with and without stratospheric aerosols, to assess the impact of stratospheric aerosol scattering and aerosol-ozone interaction on the AC process. We note that these simulations also implicitly include the similar effect of the coupling between Rayleigh scattering and aerosol absorption as the aerosols change altitude, however the volcanic aerosol is believed to be predominantly scattering[4] and thus we expect that the aerosol-ozone coupling is the dominant driver of differences. The simulations highlight the significance of the radiative interaction between these two components. If we apply the NASA standard AC algorithm[10], which assumes the aerosols are well below the ozone layer, to a signal that includes aerosol scattering contributions from within the ozone layer, the effect is to underestimate the ozone absorption losses, especially in the green-red region of the spectrum where ozone absorption is strongest (Fig. 5). Once the remainder of the atmospheric contributions to the observed signal are subtracted, the result is an underestimation of the water-leaving signal with a spectral bias that roughly follows the ozone absorption spectrum (Fig. 6). Figure 6a presents monthly ratios of retrieved Rrs($\lambda$) at 443 nm and 547 nm, as well as the derived $b_{bp}$(443) coefficient, for the SPSG region over the year 2022 relative to the average of all preceding years. Figure 6b depicts these same ratios obtained by applying standard algorithms to simulated data, where the simulations represent contrasting scenarios with and without the inclusion of stratospheric aerosols. The similarity in the seasonal trends of Fig. 6a, b strongly supports our

hypothesis that the increase in stratospheric aerosols following the Tonga eruptions introduced a pronounced and erroneous bias in the Rrs($\lambda$) retrievals for wavelengths primarily influenced by ozone, which is further propagated into erroneous trends in biogeochemical products, including phytoplankton carbon as inferred from $b_{bp}$(443).

Furthermore, due to the spectral characteristics of the Rrs($\lambda$) bias and the nature of the algorithms, all derived biogeochemical products are not affected in the same way. While both $b_{bp}$(443) and Chla are inferred from Rrs($\lambda$), the Chla algorithm, as applied over relatively clear open-ocean waters, is based on spectral band differences and is thus less sensitive to changes in Rrs($\lambda$) that affect all bands in the same direction[21]. In contrast, $b_{bp}$(443) is derived from the spectral optimization of a bio-optical model to the Rrs($\lambda$) distribution, with fixed assumptions about pure seawater contributions and the spectral shape of model components including $b_{bp}(\lambda)$, and is thus more sensitive to changes in the absolute magnitude as well as the spectral shape of Rrs($\lambda$)[22].

**Mitigation strategy**

Our findings also suggest a path forward to mitigate the impact of the Tonga eruptions and, potentially, other events that have contributed to unusual enhancements in stratospheric aerosol loads. Specifically, a series of

sensitivity analyses could be performed to characterize the effect of stratospheric aerosols as a function of concentration, altitude, and microphysical properties to produce a model or look-up table. Ancillary data from OMPS or similar sensors could then be used to compute a correction using that model, as a precursor to application of the standard AC algorithm; however, such a correction will take some time to develop and validate, and it will be limited by the availability, quality, and resolution of the ancillary aerosol information.

## Conclusions

The presence of an anomalously high stratospheric aerosol loading biases satellite OC data products. Such an anomalous stratospheric AOT following the Hunga Tonga-Hunga Ha'apai eruptions in January 2022 caused regionally and temporally covarying out-of-family behavior in OC retrieval products. These OC anomalies are seen in observations from multiple satellites, and they are not seen in in-situ metrics of phytoplankton biomass, from which we conclude that the anomalies must be due to error in the OC retrieval process. Radiative transfer simulations demonstrate that the effect of enhanced stratospheric aerosol loading is to induce biases of the type observed in the retrieved OC products, as it invalidates a key assumption in the AC algorithm on the separation of aerosol scattering and ozone absorption.

This analysis presents a caution to the OC data user community until such time as algorithms can be improved to properly compensate for enhanced stratospheric aerosols from this and other (historical and future) eruptions. Implementing such an improvement is beyond the scope of the current study and is a longer-term effort that will require accurate knowledge of the spatiotemporal variation not only of stratospheric AOT but also the aerosol microphysical properties (e.g., size, refractive index).

Looking forward, various proposals for geoengineering to counter global warming involve the deliberate injection of aerosol precursors such as $SO_2$ into the stratosphere[23,24]; indeed, major eruptions have been considered as 'natural experiments' for this type of geoengineering[25,26]. Such proposals also raise concern due to potentially severe effects on other aspects of the Earth system[25–28]. Deliberate aerosol injection would likely introduce similar biases to satellite OC and other satellite data products (e.g., sea surface temperature[29]) for the same physical reasons as observed here, which would complicate efforts to assess the true response of Earth's biosphere to anthropogenic geoengineering.

## Methods
### Ocean color data

Satellite ocean color data from six currently active missions were obtained from NASA's Ocean Biology Distributed Active Archive Center (OB.DAAC). These six missions represent three distinct sensor designs, including MODIS on both the Aqua and Terra platforms, Visible and Infrared Imaging Radiometer Suite (VIIRS) on the SNPP and NOAA-20 platforms, and OLCI on ESA's Sentinel-3A and Sentinel-3B platforms. The NASA-provided multi-mission OC data correspond to reprocessing version R2022.0 (https://oceancolor.gsfc.nasa.gov/data/reprocessing/), which utilized a common AC algorithm[10]. The acquired data products included Rrs(λ) and two products that are inferred from Rrs(λ): Chla[21,30] and $b_{bp}(443)$[22]. The $b_{bp}(443)$ is a standard NASA product that is linearly related to Cphy through a simple scale factor[31].

For all NASA satellite data, we used the global monthly (Level-3 binned) data products with a resolution of ~4.6 × 4.6 km$^2$, expressed on a sinusoidal grid, which were then spatially averaged using arithmetic means to produce regional time-series analyses. The ocean regions used in our analysis include the subpolar northern hemisphere (NH) spanning 0°–50° N latitude, the subpolar southern hemisphere (SH) spanning 0°–50° S latitude, and the South Pacific Subtropical Gyre (SPSG), a large clear-water region of the SH as defined in Signorini and McClain (2012)[32].

To evaluate any unique dependence on the NASA AC algorithm, OLCI OC time series data for the SPSG region, as processed using an independent atmospheric correction algorithm[11,33], were acquired from EUMETSAT.

When comparing OC trends across missions, results are presented using standardized anomalies from the pre-2022 annual mean value. For each sensor, the pre-2022 mean and standard deviation of monthly means are used to create a time series of anomalies quantified as standard deviations from the mean, such that a value of $+/-1$ indicates a month that is one standard deviation above or below, respectively, the pre-2022 annual mean value for that sensor. Reporting standardized anomalies removes the influence of small differences in magnitude and seasonality between the raw Rrs datasets from different missions that may arise due to differences in sampling or mean time of day, or minor calibration or algorithm errors.

### Atmospheric aerosol data

To track changes in aerosol distribution, we used NASA's OMPS limb profiler monthly stratospheric aerosol data set (available starting March 2012-present)[34]. This provides monthly mean multi-wavelength stratospheric AOT on a 5-degree latitude by 15-degree longitude grid. We use the AOT at 675 nm as representative, as retrieved AOT at shorter wavelengths is less quantitatively reliable[35]. For regional analyses, we calculated area-weighted monthly geometric means from the gridded data. We used geometric means because AOT data distributions tend to be skewed, and are thus better approximated with lognormal rather than normal distributions[36].

### In situ measurements of ocean optical properties

To evaluate in situ changes in ocean optical properties, we used measurements of $b_{bp}$ at 700 nm within the SPSG from BGC-Argo floats (https://usgodae.org/argo/argo.html), which were downloaded using the OneArgo-MAT toolbox[37]. These data were collected and made freely available by the International Argo Program and its partner organizations (https://argo.ucsd.edu, https://www.ocean-ops.org). The Argo Program is part of the Global Ocean Observing System[38]. The BGC-Argo $b_{bp}(700)$ profiles were de-spiked using a three-point moving median filter, and then aligned using the average $b_{bp}(700)$ in 700–750 m intervals. Surface data (5–20 m depth) from all floats ($n = 33$) were then grouped into 7 day intervals, and values outside of the 5$^{th}$ and 95$^{th}$ percentile were removed. The resulting 58901 data points from 2016 to 2022 were used in our regional analysis.

### Atmospheric radiative transfer modeling

To assess the expected impact on ocean color retrievals due to the unusual aerosol distributions associated with the eruptions, we utilized a vector radiative transfer model that fully couples light scattering and absorption in the atmosphere-ocean system[39,40]. We conducted simulations under two distinct atmospheric conditions. In the first scenario, we modeled an atmosphere containing a single layer of tropospheric aerosols. For the second scenario, we introduced an additional stratospheric aerosol layer, thus simulating an atmosphere with two aerosol layers. In both cases, we assume an underlying ocean surface with 5 m s$^{-1}$ wind speed and a clear water body with a Chla concentration of 0.3 mg m$^{-3}$. We account for absorbing water vapor and ozone gases in the atmosphere with concentrations of 1.25 cm and 320 Dobson units, respectively, both with the U.S. standard atmosphere 1976. The aerosol vertical distribution is modeled as a Gaussian function with a center height defined as the layer height of each aerosol layer and a layer width (one standard deviation) of 2 km[41]. The lower boundary aerosol layer (tropospheric) is assumed at 2 km height with an optical depth of 0.08, and a stratospheric sulfate type aerosol layer is assumed at 23 km with an optical depth that accounts for the monthly variations from the OMPS data averaged over the SPSG study region. These AOT values are defined at the reference wavelength of 869 nm for MODIS Aqua. We also consider the monthly variation in solar geometry at the study region and assume a fixed view zenith angle of 25° and relative azimuth of 80° to approximate the average radiant path geometries for MODIS Aqua observations of the SPSG. The boundary layer aerosols, modeled following Ahmad et al.[42], are assumed to be bimodal and coarse mode dominant with a fine mode fraction of 0.01 and relative humidity of 70%. The stratospheric aerosol, modeled following Taha et al.[4], assumes a monomodal lognormal distribution with a refractive index of 1.45 0i (purely scattering), a number median radius of 0.2 μm, and a

distribution width of 1.6 μm (standard deviation). These simulated MODIS Aqua observations are then processed through the NASA standard AC algorithm[10] to retrieve the Rrs($\lambda$), from which $b_{bp}(443)$ is then derived via the same algorithm that is used for MODIS Aqua standard products[22].

## Reporting summary

Further information on research design is available in the Nature Portfolio Reporting Summary linked to this article.

## Data availability

All data used in our analysis were acquired from public repositories. The OMPS stratospheric aerosol data is available from NASA's Goddard Earth Sciences Data and Information Services Center (GES DISC, https://disc.gsfc.nasa.gov/).The NASA ocean color data is available from the Ocean Biology Distributed Active Archive Center (OB.DAAC, http://oceancolor.gsfc.nasa.gov). The ESA/EUMETSAT ocean color time-series data from OLCI is available at https://metis.eumetsat.int/data/oc/. The BGC-ARGO in situ $b_{bp}$ data is available from https://usgodae.org/argo/argo.html.

## Code availability

The software source code used to produce all NASA ocean color data products from satellite observations is freely distributed by the OB.DAAC through the SeaDAS software package (seadas.gsfc.nasa.gov). This includes L2gen (version 9.6), which contains the atmospheric correction algorithm as well as the derived product algorithms. The same L2gen code was used here to retrieve ocean color products from simulated MODIS-Aqua data. The vector radiative transfer code used for the simulation is developed and maintained by Dr. Pengwang Zhai of the University of Maryland Baltimore County (UMBC), and is available upon request (pwzhai@umbc.edu).

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

## Acknowledgements
This work was supported through NASA's Terra and Aqua Senior Review Proposals for MODIS Algorithm Maintenance, NASA's Research Opportunities for Space and Earth Sciences Program #NNH20ZDA001N-SNPPSP, and the PACE Project. IC's contributions additionally supported NSF 222980. We are grateful to the BGC Argo community for sharing their data. We also acknowledge Dr. Pengwang Zhai of University of Maryland Baltimore County for supporting the Radiative Transfer code used in this analysis.

## Author contributions
B.F. provided the analysis of the satellite ocean color data. I.C. provided analysis of the BGC-Argo data and expertize on phytoplankton ecology. A.I. provided the sensitivity analysis to show the impact of aerosols within the ozone layer. A.S. provided analysis of the OMPS data to characterize the aerosol distribution and generated all figures in the manuscript. All authors contributed substantially to the development of the manuscript text.

## Competing interests
The authors declare no competing interests.
