## [Peer Review File · Communications Earth & Environment]

This manuscript has been previously reviewed at another Nature Portfolio journal. This document only contains reviewer comments and rebuttal letters for versions considered at Communications Earth & Environment.

5th Dec 23

Dear Mr Franz,

Thank you for transferring your manuscript titled "Anomalous trends in global ocean carbon concentrations following the 2022 eruptions of Hunga Tonga-Hunga Ha'apai" to Communications Earth & Environment. We are returning your manuscript to you with a "revise" decision so that you can make the required revisions and provide a comprehensive rebuttal to the reviewers' earlier comments and, when you are ready (ideally within 6 weeks), submit the updated files. We will then make an editorial assessment of the revisions and responses and, if appropriate, send your manuscript back to the reviewers for their assessment.

If you are able to adequately respond to these and the reviewers' other concerns, we would be happy to look at a revised manuscript; otherwise your best course of action may be to submit elsewhere.

We hope to receive your revised version as soon as possible. If you anticipate a delay of more than six weeks please let us know. We will be happy to consider your revision so long as nothing similar has been accepted for publication at Communications Earth & Environment or published elsewhere.

If you are not interested in submitting a suitably revised manuscript in the future please let us know immediately so we can close your file. If you have any questions, please contact me.

Please use the link below to submit a suitably revised manuscript and updated response to referees when they are ready.

[link redacted]

In the meantime, we regret that we remain unable to publish your current manuscript in Communications Earth & Environment. We would naturally understand, therefore, if you decided to submit your manuscript elsewhere.

We are sorry that we cannot be more positive as things stand.

Best regards,

Clare Davis, PhD
Senior Editor
Communications Earth & Environment

www.nature.com/commsenv/
@CommsEarth

Reviewer #1

Remarks to the Author:

The manuscript considers the satellite-observed anomalies in global ocean biogeochemical properties after the volcano Hunga Tonga-Hunga Ha'apai eruption in January 2022. These anomalies in the satellite ocean colour datasets from multiple satellite missions indicate a significant decrease in phytoplankton carbon concentrations. However, in-situ observations do not show these anomalies. The authors claim that a violation in the assumptions of the atmospheric correction process causes the satellite-observed anomalies in phytoplankton concentrations. The assumption violation is claimed to be due to the increased concentration of stratospheric aerosols after the eruption.

The main result, systematically biased phytoplankton carbon concentrations in satellite observations due to stratospheric aerosols, is significant as it reveals a clear issue in the satellite retrievals of phytoplankton. Therefore, the results in the manuscript help us avoid wrong conclusions about the phytoplankton concentrations based on biased satellite data. As the authors also mention, in the future, various geoengineering experiments or natural events may lead to increased concentrations of stratospheric aerosols, causing more frequent artefacts in the satellite-based phytoplankton carbon concentrations. Regardless of the important result revealing significant problems in phytoplankton carbon concentrations based on satellite data, I have major concerns about the experiments and conclusions drawn from them in the manuscript.

The authors clearly show the anomalies in satellite-based remote sensing reflectance and particulate backscattering coefficient in the Southern Hemisphere in 2022. However, to explain the anomalies, the authors only carry out a radiative transfer simulation to show that adding stratospheric aerosols causes a slight change in the satellite signal. Based on this simulation, actual satellite and in-situ observations, the authors conclude that the stratospheric aerosols are causing the bias. Based on this analysis, I am not entirely convinced that it is possible to rule out all other factors explaining the anomalies, especially as the change in the TOA reflectance is small, only 3% at maximum. For example, the radiative transfer simulation experiments do not consider the H₂O enhancement due to eruption, different vertical distribution of the aerosols, other possible gases, temperature profiles, etc.

Response: We understand the reviewer's concern, but a 3% change in TOA reflectance typically translates to a ~30% change in water-leaving reflectance, since atmospheric scattering accounts for ~90% of the observed signal at sensor, in the visible spectral range. It is a large effect. And, while it is true that the eruption injected a significant amount of water vapor into the stratosphere, the spectral bands employed by the various ocean color instruments, including those used for the atmospheric correction near 750 and 870-nm, are specifically selected to minimize the effect of H₂O absorption (and other gases). In contrast, the ozone absorption is broad-band and impacts all of the ocean color bands, with strongest impact in the blue-green. We have added Figure 5 to illustrate this point.

Further, while the stratospheric water vapor perturbation was large by the standards of the stratosphere (10%, Wilmouth et al. 2023; <https://www.pnas.org/doi/10.1073/pnas.2301994120>), the stratosphere is still very dry – about 99% of the water vapor in a typical atmosphere is within the troposphere - and so this represents a small perturbation to the total column water vapor (< 0.1%), i.e. the scattering-absorption coupling between stratospheric water vapor and aerosols or Rayleigh scattering is small.

Regarding the different vertical distribution of the aerosols, this is precisely what we are demonstrating as the cause of the anomaly in the apparent ocean color signal. Namely, there is an increase in the aerosol scattering contribution, and that additional aerosol population is located at or above the ozone layer, thus violating a fundamental assumption in the standard atmospheric correction algorithms commonly employed, which are the basis for retrieval of standard ocean color products such as bbp..

It should also be noted that the impact of aerosol height on the path reflectance observed at sensor has been shown to be negligible, unless the aerosols are strongly absorbing (e.g., Frouin et al. 2019, <https://doi.org/10.3389/feart.2019.00145>), or (as we demonstrate here) there is a change in the location of the aerosol scattering relative to the atmospheric profile of an absorbing gas.

Also, the manuscript lacks an essential step in which the authors show that the anomalous signal in stratospheric aerosols is propagated into anomalous phytoplankton carbon concentrations. The radiative transfer simulation data should be used, and the standard atmospheric correction and phytoplankton carbon concentration estimation should be carried out using the operational algorithms. This step will show if the results correspond to the ones observed by the satellites. If this test is not feasible, it should be confirmed using another analysis or explained in more detail.

Response: We thank the reviewer for this suggestion. We have extended the modeling to show impact on the phytoplankton carbon concentration, by applying the standard algorithm for bbp retrieval to the water-leaving reflectances derived from our radiative transfer simulations (new Figure 6 and associated discussion).

In line 157, the authors state that there is a strong negative deviation in $Rrs(\lambda)$ in 2022. However, in Figure 1, the deviation is much stronger at 547 nm than at 443 nm. This needs to be mentioned in the manuscript. I would also like to see an explanation why Rrs at 443 nm does not have such a large anomaly, but b_{bp} at 443 nm has.

Response: We thank the reviewer for this suggestion. In fact, the stronger impact in the green relative to the blue is consistent with the spectral signature of ozone absorption (new Figure 4), and this observation is what first suggested to us that the anomaly may be related (in some way) to ozone. We have added this observation to the manuscript.

Regarding the translation from Rrs anomaly to b_{bp} anomaly, b_{bp} is derived from the model of Werdell et al. 2012. Briefly, the Werdell model defines Rrs as being proportional to $b_w/(a+b_b)$, where a and b_b are the total water absorption and backscatter coefficients, and where a is a sum of absorption from the water (a_w), phytoplankton (a_{ph}), and detritus/gelbstoff (a_{dg}). Total backscattering (b_b) is modeled as the sum of backscattering from pure seawater (b_{bw}) and particles (b_{bp}). With a_w and b_{bw} known (Pope and Fry 1997, <https://doi.org/10.1364/AO.36.008710> and Zhang et al 2009, <https://doi.org/10.1364/OE.17.005698>), and the spectral dependence of b_{bp} , a_{ph} , and a_{dg} assumed (adopted from previously published observations, see Werdell et al, 2012 and references within), the model is optimized to retrieve the magnitude of $b_{bp}(\lambda)$, $a_{ph}(\lambda)$, and $a_{dg}(\lambda)$ that best fits the retrieved Rrs spectrum. Thus, it is the overall spectral shape of Rrs , and not the magnitude at a specific wavelength, that drives the magnitude of bbp at all wavelengths (λ). We have added to the discussion of Figure 6 to clarify this point.

I would also like to have a more detailed explanation of the delay (about 3-4 months) from the eruption

(January) to the effect seen in the Rrs data. Maybe adding a figure showing a scatter plot of the bias in OMPS stratospheric AOT against bias in some satellite R_rs would make it more clear.

Response: As shown in Figure 3, the eruption happened in Jan and the AOT peak (based on OMPS data) happened in June-July (consistent with the delayed anomaly in Rrs data). The delayed peak in AOT is due to the oxidization process that gradually converted the initial injection of SO₂ into stratospheric sulphate aerosols. This is noted in the introduction with reference to Legras et al. 2022. We have added further clarification to the discussion of Figure 3, as:

“The delayed peak in AOT is due to the oxidization process that gradually converted the initial injection of SO₂ into stratospheric sulphate aerosols. This progression to positive anomaly in the AOT ...”

We also note that, while this is an important phenomenon, it is outside the scope of this paper to further study the atmospheric processes leading up to delay in formation of stratospheric aerosols from the eruption, and we believe the clarification and current reference should offer enough explanation and direct the reader to more detailed papers on this topic.

Overall, the manuscript is well-written and easy to follow. The data sources used are widely used datasets, and their use has been well justified.

Response: Thank you.

Minor comments:

* The manuscript and even the title mention global ocean carbon concentrations. However, the manuscript’s data is not actually global, and the most detailed analysis concentrates on a specific region and the anomalies are seen only in the Southern Hemisphere. Of course, the anomalies are so significant that they also affect the global statistics, but it could be more informative to be more specific about the region mostly affected.

Response: The anomaly was first observed when looking at trends in global (deep ocean) carbon concentrations, and then breaking it down by hemisphere, as in Franz et al. 2023. The effect is readily observed through-out the southern hemisphere and into the equatorial region (Franz et al. 2023, Fig. 3.23). We appreciate the suggestion but feel the “global” moniker is appropriate where used.

* I.40. Citation style "[5]" differs from general style

* I.42. Citation style "[6]" differs from general style

Response: This was to avoid the chance of confusing a citation reference with an exponent, i.e., km⁶. We leave this to the discretion of the editor.

* I.44. The authors mention H₂O enhancement, but the experiment did not assess or comment on the effect of the H₂O enhancement.

Response: As noted and illustrated in new Figure 5, the effect of H₂O enhancement on the spectral bands used for ocean color is expected to be negligible. We direct the reviewer to our previous answer to their

general comment, and Wilmouth et al., 2023 paper.
<https://www.pnas.org/doi/10.1073/pnas.2301994120>

* I.81. The b_{bp} is playing an important role in the manuscript. Please explain in more detail to improve readers' understanding of what factors affect this term. Are there other factors besides (the linear dependency of) C_{phy} that should be considered in the analysis?

Response: Phytoplankton carbon concentration (C_{phy}) is directly computed from b_{bp} at 443nm via a simple scale factor, as derived in Graff et al. 2015. We have added this clarification. No other factors affect that relationship. As explained above in answer to the general question from the reviewer, it is the spectral shape of R_{rs} that drives the magnitude of b_{bp} (at all wavelengths), and the magnitude of b_{bp} that drives the magnitude of C_{phy} .

* I.87. "South Pacific Subtropical Gyre". It would be good to have a map or other information that clearly and uniquely describes the regions of interest used in the manuscript.

Response: We introduce the SPSG region with a reference to Signorini & McClain (2012), wherein their Figure 1 provides a global map with regional overlay. <https://doi.org/10.1080/01431161.2011.625053>. We have added "a large clear-water region of the SH as defined in Signorini and McClain (2012)" to make that explicit. We prefer not to add an additional figure for this, since it is available in the cited reference, and it is not a simple region to define with cartesian coordinates.

* I.127. What is meant by a layer width of 2 km? Is it 1std, 3std, or some other width?

Response: Yes, 1std. We have added this clarification.

* I.222. In the manuscript, mixed use of symbols and text, such as "backscatter", is used. It would be easier for the reader to follow if the authors used a consistent notation throughout the manuscript.

Response: We replaced backscatter with b_{bp} where appropriate.

* I.231. "Franz et al.", consider adding a citation to the Franz et al. (2023).

Response: The citation is provided at the end of the sentence.

* I.289. It would be beneficial if the spectral TOA reflectance coupling bias magnitude were compared with the typical instrument noise levels to show that they are significant and do not mix with the observation noise

Response: The reported trends are derived from monthly averages of large regions of the global ocean. The native resolution of the satellite instruments is $\sim 1.4 \text{ km}^2$ or higher, and the signal to noise is on the order of 2000 in the bands used for ocean color, at typical TOA radiances over ocean (Hu et al., <https://doi.org/10.1364/AO.51.006045>). Each monthly average value represents 10s of millions of observations. Assuming instrument noise is reduced by $N^{0.5}$, the contribution of instrument noise to the ocean color trends presented here is negligible.

* I.294. Please define ρ_w . Also, the figure would be easier to read if instead of " ρ_w ", a separate axis for ρ_w would be added on the right side

*Response: The ρ_w is water-leaving reflectance, which is effectively just $\pi * R_{rs}$. This confusion has been removed with the updated figures.*

Review #2

Remarks to the Author:

The paper aims to analyze anomalous trends in phytoplankton carbon concentrations following the Hunga Tonga-Hunga Ha'apai volcanic eruption in January 2022, as observed in multiple ocean color data sets from various satellite instruments. The authors conclude that this trend is attributable to retrieval errors caused by inaccurate aerosol corrections. Overall, the paper is well written and well organized, demonstrating a solid foundation for its central theme. My primary concern lies in the incomplete attempt to simulate the effects of inaccurate aerosol corrections on the observed trends. To enhance the paper's scientific rigor, it is recommended to conduct more comprehensive simulations that replicate the observed anomalous trends and explore how inaccurate aerosol corrections might contribute to them. I recommend this paper for publication once these concerns and suggestions are adequately addressed.

Response: Thank you. We accept that a more complete simulation is needed to fully support our conclusions, and we have updated the manuscript as discussed below. We believe that this addresses the reviewer's concerns, confirms our previous conclusions, and greatly improves the manuscript.

Specific comments:

L16-20: "The enhanced concentration of aerosols in the stratosphere following the eruptions results in a violation of some fundamental assumptions in the processing algorithms used to obtain marine biogeochemical properties from satellite radiometric observations, and it is demonstrated through radiative transfer simulations that this is the likely cause of the anomalous trends."

The statement is inaccurate and does not reflect the work presented in this paper. The simulations presented by the authors failed to reproduce the anomalous trends and explain the role of stratospheric aerosol in causing this anomaly. See my comment below.

Response: We have expanded the simulation and processed the simulated radiances to Rrs and bbp to show more clearly the impact of stratospheric aerosols to ocean color trends. This is presented in the new Figure 6 and associated discussion. We believe the expanded simulation results strongly support our original statements.

L38: "and a large amount of SO₂"

Change to "and a modest amount of SO₂". Compared to other eruption, the amount of SO₂ injected is small

Response: Done. It was a modest injection of SO₂ compared to some passed eruptions, though still significant compared to no eruption.

Figure 1: Besides 2022, the Rrs(λ) at 443 nm shows a couple of anomalous years in the SH. Can you comment on the causes of this anomaly?

Response: The largest negative anomalies in Rrs(443) occur in 2010 and 2019-2020. The 2010 anomaly is due to a very pronounced La Nina, where the influx of cold water and associated nutrients results in

enhanced phytoplankton productivity. Similarly, 2019-2020 is due to the massive Australian wildfires of that year, which carried nutrients to nutrient-deplete regions of the southern ocean and produced unusual phytoplankton blooms (see Tang et al. 2020, <https://doi.org/10.1038/s41586-021-03805-8>).

Notably, these two events capture actual changes in phytoplankton chlorophyll concentration, which is manifest in the ocean color signal as reduced Rrs in the blue (due to increased absorption by chlorophyll) with relatively little impact to Rrs in the green, as we would expect. Dramatic changes in Rrs in green, in relatively clear, optically deep waters far from terrigenous influences (i.e., suspended sediments) are more difficult to explain through any bio-optical processes. This was a primary reason for our suspicion that the effect was likely indicating an error in atmospheric correction.

Figure 2: The figure indicates that the anomalous trends did not end in 2022. Can the authors comment on whether they see this trend in 2023?

Response: The trends do continue into 2023, but other climatic event also play a role in what we are able to observe from ocean color alone, as 2023 marks the end of a modest but extended La Nina period and transition to El Nino conditions. It would be difficult to accurately assess (from the ocean color alone) where the influence of the eruption on the ocean color signal ends.

Figure 3: Unlike the text and caption, the figures' titles show 510 nm instead of 675 nm. Please correct the figures title.

Response: Done. Thank you. This was a mistake in the figure (the caption is correct). We had already fixed this, but neglected to update the embedded figure in the submitted draft.

L215-216: "The similarity in ocean color (Figure 1 and 2) and aerosol trends (Figure 3) over the year 2022 suggests that there is likely a causal link"

The ocean color trends are anti-correlated with the aerosol trends, not similar. Please correct the statements.

Response: Replaced "similar" with "anticorrelated" to clarify that similarity is expressed in the inverse.

L241-246: "The NASA AC algorithm assumes that aerosols are primarily scattering, with only weak absorption, and that those aerosols are located in the troposphere. When first injected, the additional aerosols from the Tonga eruptions are believed to have been moderately absorbing, but after aging and transport the sustained anomalous aerosol population in the stratosphere is found to be weakly absorbing, consistent with the NASA AC algorithm assumptions."

I do not understand this argument. Almost all published studies of this eruption agree that the main Hung-Tonga aerosol plume was between 20-26km and composed of sulfate aerosol (Taha et al., 2022; Legras et al., 2022; SelliCo et al., 2022), and any ash or absorbing particles were only observed the first day at 35 km altitude. Baron et al., 2023 reported a possible absorbing aerosol measurement at 34 km, not within the bulk of the volcanic plume. The aerosol type and properties are not the source of this error, although the larger than usual particle size of this eruption (Taha et al., 2022; Khaykin et al., 2022; Zhu et al., 2022) might be a source of uncertainty if the NASA algorithm assumes small sulfate particles. The main issue here might be

the assumption that all the aerosols are located in the troposphere, not that it is primarily scattering particles.

Response: We agree that the sustained aerosol contribution from the eruption was non-absorbing. That fact matters, because the NASA AC algorithm (and the ESA algorithm) does not account for strongly absorbing aerosols. If in fact this eruption produced an abundance of strongly absorbing aerosols, that would be another violation of our algorithm assumptions. The fact that the additional aerosols are predominately scattering is important to our hypothesis, as is the fact that they are located in the stratosphere.

L253-268: “To evaluate the second hypothesis, we assess the impact of this unusual aerosol-ozone mixing on the AC process, and”

The authors correctly identified the need for radiative transfer simulations with realistic aerosol loading in the context of the Hunga-Tonga eruption. However, their attempt to attribute the observed anomaly solely to inaccurate treatment of ozone-aerosol radiative interaction is unconvincing and inadequately supported by existing literature. Their assertion that algorithm inaccurate treatment of ozone-aerosol radiative interaction is the cause of this anomaly is weak, and not supported by the literature.

The simulation presented in Figure 5 is incomplete. It lacks essential details such as the date and location of the simulation, making it challenging for readers to contextualize the results. I also find that the authors choice to present simulations with and without ozone, rather than with and without stratospheric aerosol perplexing. At best, they only succeed to demonstrate why ozone corrections are needed for ocean color retrievals. To demonstrate the need for a more accurate account of the aerosol plume in the stratosphere, it would have been more appropriate to compare simulations with and without the stratospheric aerosol.

The paper misses an opportunity to provide a more comprehensive simulation that could guide efforts to mitigate the impact of the Hunga-Tonga eruption. A more effective approach would have involved conducting full simulations that reproduce the anomalous Rrs, as seen in the work by Jia et al. (2023).

I suggest that the authors conduct a more extensive literature review to support the claim that ozone-aerosol radiative interaction is the key factor contributing to the observed anomaly and repeat the simulations with and without stratospheric aerosol, which is more relevant to the research question and more likely to yield meaningful insights. The authors should also consider conducting more comprehensive simulations that closely replicate the anomalous Rrs observed, to guide mitigation efforts effectively.

Response: The simulation as presented in the submitted draft is in fact a comparison of “simulations with and without the stratospheric aerosol”. That was perhaps not clear in the wording of the figure caption, but that’s what was meant by “with and without coupling the ozone-aerosol interaction”. We put the aerosols in the ozone layer such that coupling occurred.

We agree, however, that a more complete simulation is needed to better support our conclusions. To that end we simulated the MODIS Aqua radiances at the top of the atmosphere over the SPSG region, for monthly observations spanning the year 2022. We simulated the data with and without the addition of stratospheric aerosols, where the temporal distribution and microphysical properties of the stratospheric

aerosols were selected to mimic what has been reported in the literature and inferred from OMPS data. We processed these two cases through the standard AC algorithm, and show that the impact on Rrs and the derived bbp(443) closely follow the effects observed in the satellite data from 2022 (relative to the historical norm). We feel this more comprehensive analysis strongly supports our original conclusions, and greatly enhances the quality of the manuscript.

29th Feb 24

Dear Mr Franz,

Your manuscript titled "Anomalous trends in global ocean carbon concentrations following the 2022 eruptions of Hunga Tonga-Hunga Ha'apai" has now been seen by one of the original reviewers and a new replacement reviewer (Reviewer #3), and we include their comments at the end of this message. They find your work of interest, but Reviewer #3 raises some important points. We are interested in the possibility of publishing your study in Communications Earth & Environment, but would like to consider your responses to these concerns and assess a revised manuscript before we make a final decision on publication.

We therefore invite you to revise and resubmit your manuscript, along with a point-by-point response that takes into account the points raised. Please highlight all changes in the manuscript text file.

Please use the following link to submit your revised manuscript, point-by-point response to the referees' comments (which should be in a separate document to any cover letter), a tracked-changes version of the manuscript (as a PDF file) and the completed checklist:

[link redacted]

We hope to receive your revised paper within six weeks; please let us know if you aren't able to submit it within this time so that we can discuss how best to proceed. If we don't hear from you, and the revision process takes significantly longer, we may close your file. In this event, we will still be happy to reconsider your paper at a later date, as long as nothing similar has been accepted for publication at Communications Earth & Environment or published elsewhere in the meantime.

Please do not hesitate to contact us if you have any questions or would like to discuss these revisions further. We look forward to seeing the revised manuscript and thank you for the opportunity to review your work.

Best regards,

Clare Davis, PhD
Senior Editor
Communications Earth & Environment

www.nature.com/commsenv/
@CommsEarth

EDITORIAL POLICIES AND FORMATTING

Editorial Policy: Policy requirements (Download the link to your computer as a PDF.)

Furthermore, please align your manuscript with our format requirements, which are summarized on the following checklist:

Communications Earth & Environment formatting checklist

and also in our style and formatting guide Communications Earth & Environment formatting guide .

*** DATA: Communications Earth & Environment endorses the principles of the Enabling FAIR data project (<http://www.copdess.org/enabling-fair-data-project/>). We ask authors to make the data that support their conclusions available in permanent, publically accessible data repositories. (Please contact the editor if you are unable to make your data available).

All Communications Earth & Environment manuscripts must include a section titled "Data Availability" at the end of the Methods section or main text (if no Methods). More information on this policy, is available at <http://www.nature.com/authors/policies/data/data-availability-statements-data-citations.pdf>.

If a community resource is unavailable, data can be submitted to generalist repositories such as figshare or Dryad Digital Repository. Please provide a unique identifier for the data (for example a DOI or a permanent URL) in the data availability statement, if possible. If the repository does not provide identifiers, we encourage authors to supply the search terms that will return the data. For data that have been obtained from publically available sources, please provide a URL and the specific data product name in the data availability statement. Data with a DOI should be further cited in the methods reference section.

REVIEWER COMMENTS:

Reviewer #1 (Remarks to the Author):

I reviewed the original version of the manuscript. In this revised manuscript, the authors have successfully addressed all my comments and improved the manuscript. The manuscript now provide a robust evidence that inaccurate aerosol correction caused the ocean color data, and more specifically the phytoplankton carbon concentrations, trends that were not real. I have no further comments.

Reviewer #3 (Remarks to the Author):

This paper reports on anomalies in retrieved particle backscattering coefficient from space-borne ocean color sensor data for the year 2022 when compared to the historical record. These anomalies were observed only in the Southern Hemisphere (SH) retrievals and were not observed in in-situ data for a region (in the SH) where such data were available. Their attribution of the anomalies to errors in atmospheric correction induced by the injection of stratospheric aerosols by the volcano Hunga Tonga-Hunga Ha'apai in January 2022 is the central focus of the paper.

The paper is well written, and the authors make a strong case for their assertion through end-to-end simulations using published properties of the stratospheric aerosol. These simulations mimic the observed anomalies (Figure 6). I recommend publication; however, I would like to raise a couple of points for the authors to consider and possibly to respond with comments in the final version of the paper.

First: Figure 2 panels (e) -- (h) suggest considerably different seasonal trends in Rrs in the green prior to 2022 depending on which algorithm was used in processing the sensor data (NASA versus ESA). The NASA algorithm applied to the EUMETSAT OLCI data yields negative anomalies in the June-July time period that are similar to those observed with MODIS, while the ESA algorithm applied to the same data yields neutral or even positive anomalies. In fact, it appears that the differences induced by using different algorithms are nearly as large as the observed anomalies that are the focus of this study. Do the authors have any explanation for this behavior? As I understand, the principal difference in approach between the two algorithms is that NASA's places all the aerosol (and its variation) in the marine boundary-layer, while in the ESA algorithm the free troposphere and stratosphere each contain a fixed amount of aerosol, in addition to an aerosol of variable concentration (and type) in the marine boundary-layer.

Second: the authors seem to suggest that the main impact of the stratospheric aerosol is through an enhancement of Ozone absorption. However, examination of Figure 6 reveals that the stratospheric aerosol's effect at 547 nm, where there is significant Ozone absorption, is only about 50% more than the effect at 443 nm, where there is negligible Ozone absorption. This suggests that the algorithm error at 443 nm has more to do with the vertical distribution of the aerosol, i.e., the neglect of aerosol vertical structure in the NASA algorithm, and that at 547 nm both aerosol vertical structure and aerosol-Ozone interaction are likely important. The Rayleigh-aerosol interaction is likely to be significantly different when the aerosol is distributed vertically compared to the case (usually assumed) where the aerosol is all in the marine boundary layer.

Reviewer #1

Remarks to the Author:

I reviewed the original version of the manuscript. In this revised manuscript, the authors have successfully addressed all my comments and improved the manuscript. The manuscript now provide a robust evidence that inaccurate aerosol correction caused the ocean color data, and more specifically the phytoplankton carbon concentrations, trends that were not real. I have no further comments.

Response: We thank you for the original review. We feel it has greatly improved the manuscript.

Reviewer #3

Remarks to the Author:

This paper reports on anomalies in retrieved particle backscattering coefficient from space-borne ocean color sensor data for the year 2022 when compared to the historical record. These anomalies were observed only in the Southern Hemisphere (SH) retrievals and were not observed in in-situ data for a region (in the SH) where such data were available. Their attribution of the anomalies to errors in atmospheric correction induced by the injection of stratospheric aerosols by the volcano Hunga Tonga-Hunga Ha'apai in January 2022 is the central focus of the paper.

The paper is well written, and the authors make a strong case for their assertion through end-to-end simulations using published properties of the stratospheric aerosol. These simulations mimic the observed anomalies (Figure 6). I recommend publication; however, I would like to raise a couple of points for the authors to consider and possibly to respond with comments in the final version of the paper.

First: Figure 2 panels (e) -- (h) suggest considerably different seasonal trends in Rrs in the green prior to 2022 depending on which algorithm was used in processing the sensor data (NASA versus ESA). The NASA algorithm applied to the EUMETSAT OLCI data yields negative anomalies in the June-July time period that are similar to those observed with MODIS, while the ESA algorithm applied to the same data yields neutral or even positive anomalies. In fact, it appears that the differences induced by using different algorithms are nearly as large as the observed anomalies that are the focus of this study. Do the authors have any explanation for this behavior? As I understand, the principal difference in approach between the two algorithms is that NASA's places all the aerosol (and its variation) in the marine boundary-layer, while in the ESA algorithm the free troposphere and stratosphere each contain a fixed amount of aerosol, in addition to an aerosol of variable concentration (and type) in the marine boundary-layer.

Response: We thank the reviewer for this comment. We have noted the significant differences between the NASA and ESA processed OLCI data and concur with the reviewer that differences in atmospheric correction approaches are likely the major contributor here. No satellite sensor to date has sufficient information content to unambiguously retrieve all relevant geophysical parameters and so simplifying assumptions are always necessary. This in turn can manifest in different regional and seasonal biases between different processing approaches.

We have confirmed with the data provider (EUMETSAT, Ewa Kwiatkowska) that their timeseries data was processed using the standard MERIS/OLCI algorithm of Antoine and Morel (1999), but with many changes highlighted in Zibordi et al. 2022 (<https://doi.org/10.1016/j.rse.2022.112911>). We have added that reference. A detailed analysis of the AC-algorithm differences is beyond the scope of this paper, but is part of ongoing work being done by members of our team (e.g. Bisson et al., 2021 <https://opg.optica.org/ao/fulltext.cfm?uri=ao-60-23-6978&id=456022>).

We are able to demonstrate here that both NASA and ESA/Eumetsat ocean color retrievals in the southern hemisphere for the year 2022 were significantly out-of-family compared to their longer-term climatologies, and that is the primary message for this analysis. In the NASA case, we demonstrate quantitatively that these anomalies are consistent with expected biases induced by stratospheric aerosol injection from the HTHH eruptions in January 2022. We suspect that ESA results are similarly influenced although the differences between atmospheric correction approaches mean the manifestation in ocean color retrievals is different.

Second: the authors seem to suggest that the main impact of the stratospheric aerosol is through an enhancement of Ozone absorption. However, examination of Figure 6 reveals that the stratospheric aerosol's effect at 547 nm, where there is significant Ozone absorption, is only about 50% more than the effect at 443 nm, where there is negligible Ozone absorption. This suggests that the algorithm error at 443 nm has more to do with the vertical distribution of the aerosol, i.e., the neglect of aerosol vertical structure in the NASA algorithm, and that at 547 nm both aerosol vertical structure and aerosol-Ozone interaction are likely important. The Rayleigh-aerosol interaction is likely to be significantly different when the aerosol is distributed vertically compared to the case (usually assumed) where the aerosol is all in the marine boundary layer.

Response: We thank the reviewer for this suggestion. We agree that both factors could be relevant here, and they were both considered in the simulation, but the magnitude of the Rayleigh-aerosol coupling depends on the level of aerosol absorption, and the Hunga Tonga aerosols are not believed to be strongly absorbing. We have added a sentence to clarify this point.

“We note that these simulations also implicitly include the similar effect of the coupling between Rayleigh scattering and aerosol absorption as the aerosols change altitude, however the volcanic aerosol is believed to be predominantly scattering⁴ and thus we expect that the aerosol-ozone coupling is the dominant driver of differences.”

2nd Apr 24

Dear Mr Franz,

Your manuscript titled "Anomalous trends in global ocean carbon concentrations following the 2022 eruptions of Hunga Tonga-Hunga Ha'apai" has now been seen by our reviewers, whose comments appear below. In light of their advice we are delighted to say that we are happy, in principle, to publish a suitably revised version in Communications Earth & Environment under the open access CC BY license (Creative Commons Attribution v4.0 International License).

We therefore invite you to revise your paper one last time to comply with our format requirements and to maximise the accessibility and therefore the impact of your work.

EDITORIAL REQUESTS:

*****Please take care to match our formatting and policy requirements. We will check revised manuscript and return manuscripts that do not comply. Such requests will lead to delays. *****

SUBMISSION INFORMATION:

OPEN ACCESS:

Communications Earth & Environment is a fully open access journal. Articles are made freely accessible on publication under a CC BY license (Creative Commons Attribution 4.0 International License). This license allows maximum dissemination and re-use of open access materials and is preferred by many research funding bodies.

For further information about article processing charges, open access funding, and advice and support from Nature Research, please visit <https://www.nature.com/commsenv/article-processing-charges>

At acceptance, you will be provided with instructions for completing this CC BY license on behalf of all authors. This grants us the necessary permissions to publish your paper. Additionally, you will be asked to declare that all required third party permissions have been obtained, and to provide billing

information in order to pay the article-processing charge (APC).

[link redacted]

Best regards,

Clare Davis, PhD
Senior Editor
Communications Earth & Environment

www.nature.com/commsenv/
@CommsEarth